# Can We Use the Oculus Quest VR Headset and Controllers to Reliably Assess Balance Stability?

**DOI:** 10.3390/diagnostics12061409

**Published:** 2022-06-07

**Authors:** Cathy M. Craig, James Stafford, Anastasiia Egorova, Carla McCabe, Mark Matthews

**Affiliations:** 1School of Psychology, Ulster University, Coleraine BT52 1SL, UK; 2School of Psychology, Queen’s University Belfast, Belfast BT7 1NN, UK; j.stafford@incisiv.tech; 3School of Maths & Physics, Queen’s University Belfast, Belfast BT7 1NN, UK; aegorova01@qub.ac.uk; 4School of Sport, Ulster University, Belfast BT15 1ED, UK; c.mccabe@ulster.ac.uk (C.M.); m.matthews@ulster.ac.uk (M.M.)

**Keywords:** balance assessment, VR, postural control, low-cost, visual field manipulation

## Abstract

Balance is the foundation upon which all other motor skills are built. Indeed, many neurological diseases and injuries often present clinically with deficits in balance control. With recent advances in virtual reality (VR) hardware bringing low-cost headsets into the mainstream market, the question remains as to whether this technology could be used in a clinical context to assess balance. We compared the head tracking performance of a low-cost VR headset (Oculus Quest) with a gold standard motion tracking system (Qualisys). We then compared the recorded head sway with the center of pressure (COP) measures collected from a force platform in different stances and different visual field manipulations. Firstly, our analysis showed that there was an excellent correspondence between the two different head movement signals (ICCs > 0.99) with minimal differences in terms of accuracy (<5 mm error). Secondly, we found that head sway mapped onto COP measures more strongly when the participant adopted a Tandem stance during balance assessment. Finally, using the power of virtual reality to manipulate the visual input to the brain, we showed how the Oculus Quest can reliably detect changes in postural control as a result of different types of visual field manipulations. Given the high levels of accuracy of the motion tracking of the Oculus Quest headset, along with the strong relationship with the COP and ability to manipulate the visual field, the Oculus Quest makes an exciting alternative to traditional lab-based balance assessments.

## 1. Introduction

Maintaining balance is a complex process that requires sensory inputs from the visual, vestibular, and proprioceptive sensory systems of the body. All these systems work seamlessly together to give us our sense of balance [1,2]. To maintain balance, a person must continually monitor multiple sources of information coming from different sensory inputs and continually perform the necessary adjustments to position the body and limbs, so the center of mass is in a position of equilibrium. In fact, good control of the balance system is the foundation upon which other movements are built. For instance, standing on our tip toes to place a book on a high shelf or running to intercept an opponent’s pass in sport, not only involves the control of the movement of limbs but also the control of the center of mass as the limbs move.

This ability to maintain good balance (postural control) is vital for simple everyday actions, but also for the fluid, dynamic movements needed for any type of skillful action [1,2]. Indeed, difficulties in being able to control posture appropriately are often indicative of an underlying medical condition. As a result, it has become common for scientists and clinical experts to want to assess balance abilities to determine the extent of any underlying neurological problems post head injury [3], identify opportunities for motor development [4], but also profile a person’s susceptibility to future risk (e.g., falls in older adults) [5].

Given the importance of identifying deficiencies in balance, there is a strong need to develop accessible, low-cost, valid ways of assessing postural stability. The Balance Error Scoring System (BESS) is an example of a simple balance test that is widely used in professional sport and clinical settings [6]. It uses four different types of stances (double, single dominant, single non-dominant, tandem) to modify the base of support and challenge a participant’s postural stability as they try to stand as still as possible for a fixed period (usually 20 s). During that time, a clinician/experimenter observes the participant’s posture with errors being counted (i.e., losing balance, use of arms, stepping out, etc.) and scored (maximum 10 allowed). To further assess balance, modifying visual input (i.e., eyes open, eyes closed) can be added to the BESS to increase the difficulty and complexity of balance assessment. Although the test is quick and easy to administer, it has been criticized for its lack of reliability due to the subjectivity of human raters in counting errors [7]. Furthermore, the magnitude of the error (or loss of balance) cannot be quantified, meaning the extent of the balance deficits are not measured with any level of granularity.

Another limitation of observational balance tests such as the BESS is that they lack the required sensitivity for long-term monitoring of postural control. For instance, it is thought that balance disturbances following a neurological event, such as a concussion, typically resolve within 72 h of initial injury when assessed through observation alone. However, when the same participants are assessed using objective data collection methods such as motion capture and force plates, balance disturbances can be observed up to 30 days after the initial injury [8]. The inability of the BESS to pick up these residual balance deficits following head injury may put the player at increased risk of other injuries further down the line.

Full-body motion tracking, using optical-based camera systems, has also been extensively used in postural studies when the experimental protocol makes the use of force plates impossible or inconvenient (for example when the movement area was greater than the force plate’s dimensions). In such cases, motion tracking several joints is used to accurately estimate the position of the body’s Center of Gravity (CoG). This kinematic method of measuring balance is based on the definition of the CoG, which is the imaginary point around which the force of gravity appears to act, and the combined mass of the body is concentrated. From this definition, the position of a body’s CoG can be computed as the weighted average of the position of the Center of Mass of all body segments [9]. This requires accurate anthropometric and kinematic data from all body segments and is very labor intensive [10].

Several different full-body kinematic models have been used by research teams to measure CoG movement [11,12]. While the Ground Reaction Force (GRF) double-integration method and the segmental kinematic method are the gold standard for accurate measurements of human body CoG movement, more accessible measurement methods have been shown to be viable for research too. Studies have shown that measuring the movement of the sacrum, through optical tracking or inertial sensors, also provides a usable approximation of participants’ CoG [13].

Although this method is useful, another more widely adopted approach is to measure the position and displacement of the center of pressure (COP) as a person stands still on a force platform [1,2,14]. The force platform provides a point projection of the vertical reaction forces that are represented in the anterior–posterior (AP) and medial–lateral (ML) axes of movement. Although the person may be standing still, the position of the COP will change over time as the person controls their balance through micromovements of the body that are inherent in this closed-loop feedback system (predominantly ankles (AP axis) and adductors/abductors (ML axis)).

The problem with these methods is that they often involve expensive, lab-based equipment that lacks portability and applicability to more general settings. Other alternatives that use more simple methods are now being explored. For example, a recent study showed that different visual images invoked similar changes in postural control when head position (measured using an overhead webcam) and COP measures were recorded at the same time [15]. Although the purpose of this study was to measure behavioral responses to different emotive images, the strong link between head sway and COP (correlations of 0.82 AP axis and 0.73 ML axis) means that tracking head movement could offer a promising alternative to force platforms when assessing changes in postural control. Given these findings, it is possible that a new way of assessing balance in a low-cost, reliable way could be provided by a virtual reality (VR) head mounted display (HMD). This technology would not only track head movements to capture changes in postural control but would also provide the option of manipulating a participant’s visual field using a “swinging room” style protocol that in turn would invoke change in balance stability [16]. This moving room simulation is important in balance assessment [17], as it allows the clinician to go beyond a binary eyes-open and eyes-closed manipulation and probe possible causes of balance deficits in more depth.

This study will investigate the usability of head sway measures, recorded from a low-cost, consumer-based VR headset (Oculus Quest) [18] as a means of assessing changes in postural stability. The first two parts of the study focus on the technical validation of the technology, while the third part examines its use as a means of capturing changes in postural control, induced by visual field manipulations, in a group of young, healthy adult males. The aims of the study were threefold:(1)Measure the technical reliability of the Oculus Quest to track head movement (sway) by comparing it to a gold standard motion tracking system (Qualisys, Göteborg, Sweden).(2)Measure the strength of the relationship between head sway (head movements captured by both the Oculus and the Qualisys motion capture system) and Centre of Pressure (force platform, Kistler Instruments, Winterthur, Switzerland) when performing a modified version of a balance test (BESS).(3)Determine the responsiveness of a low-cost VR headset (i) to presentation of different visual field manipulations (static and dynamic) that invoke changes in postural control when a person is in a dominant and a non-dominant stance and (ii) for measuring the resulting postural adjustments (head and hand movement) in a group of healthy adults and measure their reliability across two different testing sessions.

## 2. Materials and Methods

### 2.1. Technical Validation—Balance Tracking Hardware

A passive, reflective marker was attached in the center of a VR headset (Oculus Quest, Facebook Technologies Ltd., Menlo Park, CA, USA). Fifteen Qualisys infrared motion capture cameras (Oqus 100, Qualisys, Göteborg, Sweden) recorded the movement (*x*, *y*, *z*) of the marker attached to the VR headset at 50 Hz. Centre of Pressure was also recorded at 50 Hz using the force plate Kistler 9260 AA force platform (Kistler Instruments, Winterthur, Switzerland). Both the Qualisys and Kistler data were recorded using the same software and synchronized using the same time stamp. The Oculus Quest head movement data (*x*, *y*, *z*) were also captured at 50 Hz using a data collector function in Unity game design software (MOViR, INCISIV Ltd., Belfast, UK). This data collector also collected the visual field data that indicated the transition from one visual field to another. This was important for the analysis of visual field induced differences in postural control which represented the third part of the experiment.

### 2.2. Data Collection

The data for the technical validation were obtained using a single subject design in which the female volunteer (height 1.68 m; mass 59 kg) provided informed consent and was noted as right foot dominant. The participant was asked to remove her shoes and stand on the force platform wearing the Oculus HMD and hold the two hand controllers. For all conditions, the participant was immersed in a virtual gym where two red spheres were positioned in front of the participant at elbow height. The participant was asked to place the controllers inside the spheres, so they turned green. This was to encourage the participant to keep the arms as still as possible while performing the balance tests. The Oculus Quest was calibrated using its own built-in “Guardian setup” procedure so that the Anterior/Posterior (AP, *X* axis) and Medial Lateral (ML, *Y* axis) axes were aligned with those of the force plate. The *Z* axis for all systems represented the vertical axis. The Qualisys system was also calibrated in alignment to the Oculus Quest defined axes and the calibration residual was deemed acceptable (<0.80 mm). Data collected from the Oculus and the Qualisys motion capture system were synchronized using two easily recognizable events (small vertical jump and oscillatory head movements) that were performed by the participant at the start of each trial. These two discrete events were used in the analysis to temporally align the Qualisys and Oculus head movement time series data.

### 2.3. Technical Validation—Testing Conditions

The participant performed a modified version of the BESS test that involved standing as still as possible in three different stances—double, tandem, and single leg dominant (30 s each stance). If the participant lost balance or moved out of the instructed stance, she was told to regain her balance and assume the instructed stance as soon as possible. The 3 stances were presented in the following order:Double: Standing with both feet side by side.Tandem: Standing with the heel of the right foot just in front of the toe of the left foot (non-dominant foot at the back).Single leg dominant: Standing on her right foot, hip flexed to approximately 30° and left knee bent upwards to approximately 45°.

The participant performed 3 consecutive sets of testing for each of the stances giving 13,500 data points per signal for analysis.

### 2.4. Behavioural Validation—Visual Field Manipulation

To see if virtual reality can also be used to reliably manipulate the visual field and induce measurable changes in postural control, a group of 30 healthy adult males (mean age = 26.3 ± 5 years; mean height = 1.82 ± 0.06 m); mean weight = 85.1 ± 7.6 kg; right foot dominant *n* = 23) gave informed consent and agreed to participate in the experiment.

Testing took place in a room with a solid floor and participants were asked to remove their footwear. The Oculus Quest was placed on the participant’s head and participants were asked to hold the hand controllers in their hands. The environment presented in the headset was a virtual gym. Once the participant was familiar with the environment and comfortable, they were asked to adopt a tandem stance with their left foot in front of their right foot (see Figure 1). A tandem stance was selected as it demonstrated the best relationship between the measures of COP and head sway in the technical validation study (see Section 3). Participants were asked to place the hand controllers in a standardized position (virtual spheres at elbow height in the virtual gym) and stand as still as possible until the trial ended. Each trial lasted 40 s with participants given a short break before performing the trial again with the opposite leg forward. During each trial the visual field was manipulated to create different sensory processing demands to challenge postural control. These visual field manipulations were created using off the shelf VR software (MOViR; INCISIV Ltd.) programmed using Unity3D and presented inside the headset at 2880 × 1600 pixel resolution (1440 × 1600 pixels per eye) with a 90° field of view and a refresh rate of 90 Hz.

The changes in the visual field were classed as either static or dynamic and lasted for 10 s. The static visual field had two conditions: (i) Static Light—a stationary well-lit virtual gym, and (ii) Static Dark—a stationary dark virtual gym. These two static visual fields replicated the traditional ‘eyes open’ or ‘eyes closed’ balance tests, respectively. The dynamic visual field had two conditions: (i) Dynamic Forward/Back—a virtual gym that simulated forward-backwards room motion (in the anterior posterior axis), and (ii) Dynamic Tilt—a virtual gym with rotational (tilt) movement around the anterior posterior axis. The forward–backwards manipulation involved the virtual gym moving away from the participant (2 cm/s) for 5 s, and then back towards the participant at 2 cm/s for 5 s. The rotation (‘tilt’) of the anterior–posterior axis consisted of positive roll for the 1st 5 s and negative roll for the last 5 s at a rate of 5 degrees/second. Participants completed all 4 conditions, for tandem stance with one trial for the left and one trial for the right foot forward, on two separate occasions (4 days apart to test reliability of measures).

### 2.5. Technical Validation—Data Analysis

Both the head movement captured using Qualisys motion capture technology and the Oculus Quest were recorded at 50 Hz in three axes of motion (*x*, *y*, *z*). The total distance travelled (head sway) was calculated in all three axes, whereas the distance covered by the COP was calculated only in the anterior posterior (*y*) and medial lateral (*x*) axes, as per the recognized method [10]. Intraclass correlations (ICC) were used to determine the level of correspondence between the different signals. ICCs were calculated using the standard method and values interpreted as follows: <0.4 = poor; 0.4–0.59 = moderate; 0.6–0.79 = good; >0.8 = excellent [19]. Root Mean Square Errors (RMSE) were calculated in millimeters to determine the level of precision of head movement captured using the Qualisys motion capture system and the Oculus Quest.

### 2.6. Behavioral Validation—Data Analysis

The postural changes induced by the visual field manipulations were measured using the head and hand controllers of the Oculus Quest. The head and hand controller data were captured with the same time stamp as the visual field manipulations so that the total sway for each visual field manipulation could be analyzed separately. The test was repeated 4 days later so that the reliability of the measures could also be tested using an analysis of measurement variance.

## 3. Results

### 3.1. Technical Validation Part 1—Reliability of Head Movement Captured Using the Oculus Quest

The first part of the analysis aimed to see how closely the head movement captured using the Qualisys motion capture system corresponded to the data captured from the Oculus Quest’s proprietary in built motion sensing system. Head movement data captured using the Qualisys system were directly compared with that obtained using the Oculus Quest HMD. The graphs in Figure 2 show how closely the two signals correspond when the participant adopted three different stances (Double, Tandem and Single). As expected, the most sway (movement) took place in the Single leg stance with the least in the Double leg stance condition and the Tandem stance in between.

In terms of the similarity of the signals, both the Intraclass Correlation Coefficients (ICC) and Root Mean Square Errors (RMSE) were calculated for the head movement data. As can be seen from Table 1 the Tandem and the Single leg stances had the highest ICC values (≥0.99) whereas the Double leg stance was slightly lower (0.936) for the distance (sway) calculations. In terms of the RMSE values, both the Double and Tandem stances had the lowest RMSEs (3.8 mm and 3.9 mm, respectively) with the Single leg stance being slightly higher (4.7 mm). Overall, these values indicate a very strong correspondence between the two signals and a very high level of accuracy for the Oculus Quest head movement data.

### 3.2. Technical Validation Part 2—Comparing COP and Head Sway as Measures of Postural Control

The second part of the analysis aimed to see if head sway (distance), recorded from the Qualisys and the Oculus Quest, can be compared to the COP measured using a Kistler force platform. In this analysis the three different stances (Double, Tandem, Single) were analyzed separately. The Intraclass Correlation Coefficients (ICC) were calculated for each data set (*n* = 1500 per set, and three sets per stance) to determine the degree of similarity between the measures of COP and head movement. Figure 3 shows that although the COP and head movement are capturing data at extreme ends of the body, both measures appear to reflect similar changes in postural control.

The Intraclass Correlation Coefficients showing the relationship between Head Sway (Oculus and Qualisys) and COP excursion in each of the axes (ML and AP) for the three different stances can be found in Table 2. Very strong ICC values were found for the Tandem stance when head sway distance was calculated using the Oculus (0.888) and the Qualisys motion capture data (0.879). The values for Double (0.546 and 0.451) and Single leg stances (0.654 and 0.643) were considerably lower, indicating a less strong relationship between the two measurements.

### 3.3. Measuring the Effects of Visual Field Manipulation on Postural Control

The final part of the study looked to (i) measure the effects of four visual field conditions and two types of tandem stance (dominant vs. non-dominant) on postural control in a group of 30 healthy, adult males’ and (ii) check to see if the measures of postural control were reliable over two different testing sessions. The similarity between the sway measures captured 4 days apart using the Oculus Quest was found to be very strong (r = 0.84; *p* < 0.0001). This affirmed that the balance measures were reliable across testing sessions. As a consequence, the average of the sway measures from the two sessions were used for all subsequent analyses. A two-way ANCOVA showed that there was indeed a significant main effect of visual field manipulation (F_(3,499)_ = 73.7; *p* < 0.0001) on mean total sway (across the two sessions) that was moderated by stance type (dominant vs. non-dominant) (F_(1,499)_ = 9.6; *p* = 0.002). Post hoc analysis showed how the Static Light condition had significantly less sway than the Static Dark but also than the Dynamic Tilt conditions (see Figure 4) (*p* < 0.001). Although the Dynamic Forward-Back condition (mean = 31.4 cm) was marginally better than the Static Light condition (mean = 34.0 cm) this difference was not significant (*p* = 0.685). It was, however, significantly better than both the Static Dark (mean = 61.7 cm; *p* < 0.001) and the Dynamic Tilt conditions (mean = 82.8 cm; *p* < 0.001). These differences can be explained by the Dynamic Tilt manipulation forcing postural corrections in the medial lateral axes, the axis that is least stable when a participant adopts a Tandem stance where the base of support is at its narrowest in the medial lateral axis.

In terms of effects of stance on postural control, Figure 4 shows the mean average total sway when the visual manipulations occurred in the dominant stance compared to the non-dominant stance.

## 4. Discussion

This study aimed to see if a low-cost Virtual Reality technology (Oculus Quest) could reliably assess balance. The first part looked at the technical aspects of the technology and assessed how accurately the Oculus Quest could track head movements during a simple balance task (modified version of the BESS) compared to the Qualisys motion capture system. The data clearly showed that the level of correspondence between the Oculus Quest head movement recordings and that of the Qualisys motion capture system were almost perfect (ICC > 0.99) with less than 5 mm error difference between the two signals (*n* = 4500). This was in line with a similar study that showed the Oculus Quest was able to track a user’s head movement with a mean positional accuracy of 6.86 mm [20,21].

Although this high degree of accuracy and reliability supports the potential use of VR as a viable head tracking technology, it was important to see if head movement (head sway) captures changes in postural control in a similar way to other balance measurement systems. The second technical part of the study looked at the correspondence between balance control measured using head sway (measured by the Oculus Quest and the Qualisys motion capture system) and balance control measured simultaneously using the Centre of Pressure data captured using a Kistler force platform. Although a perfect relationship is not expected as the COP is measuring what is happening between the feet and the ground and head sway is capturing head adjustments, it was predicted that both signals are capturing essential elements of postural control that will be strongly related.

The data showed that the type of stance influenced the ICC values (comparisons between COP measures and head sway measures), with the Double stance showing a moderate relationship (0.55 (Oculus); 0.45 (Qualisys)), Single stance showing a good relationship (0.65 (Oculus) and 0.64 (Qualisys) and Tandem stance showing an excellent relationship (0.89 (Oculus) and 0.88 (Qualisys) when it comes to comparing head movement distance and COP excursion. The lower correspondence between the Double stance can be explained by the fact subtle micromovements can be made between the feet and the ground to change the distribution of weight across this wider base of support. In this case, movements of the head to adjust postural imbalance is likely to be minimal. The opposite is true for the Single stance condition where there is only one foot in contact with the ground and the registration of postural adjustments through the COP are less obvious as the other limbs and the head will be used to control balance. Tandem stance, however, is still a double support stance meaning more force will be exerted through the ground giving more credence to the COP measure. Importantly, however, in a Tandem stance (see Figure 1), the base of support is narrowed in the medial-lateral axis meaning any movements to adjust posture will be in the medial–lateral axis and will be accentuated through movements of the head. This makes head sway an ideal candidate for measuring changes in postural control in Tandem stance. Furthermore, the Oculus head measurements were ever so marginally stronger than Qualisys, again offering support for low-cost VR head tracking as an alternative to more expensive balance measurement systems.

The final part of the study introduced factors that are known to challenge the balance system, namely the type of support (dominant versus non-dominant) and visual field manipulations (similar to the ‘moving room’ paradigm [17]) and tested the effects on postural control at two different time points (4 days apart). The predictions were that postural control would be best (minimal total sway) for the Tandem balance dominant foot conditions compared to the non-dominant conditions, and in the Static Light and Dynamic Forward/back conditions compared to the Static Dark and Dynamic Tilt conditions [20]. We also predicted that the measures would be stable over time with a strong relationship between the measures when participants were re-tested using the same stances and visual field manipulations 4 days later. Our analysis of the balance control exhibited by a group of thirty healthy adult males showed that there was excellent reliability in the measures of sway across two different testing sessions (r = 0.84), which affirmed that the test–retest reliability of these types of balance tests was excellent. The analysis of the sway in the different visual field conditions showed that the Static Light and Dynamic Forward/Back conditions yielded the least amount of postural adjustment. On the other hand, Static Dark and Dynamic Tilt both induced significant postural adjustments that were captured by the measures of sway (*p* < 0.001). This was in line with previous literature that showed that eyes closed and tilt conditions [20] perturb balance the most. The effect of the Dynamic Tilt condition on sway are more pronounced in Tandem stance as corrective postural movement adjustments are mainly in the medial lateral axis where the base of support was narrowest. The effect of stance, dominant versus non-dominant, also significantly affected postural control, with sway being significantly less in the dominant stance condition.

In short, this study provides strong evidence that VR technology can be used as an accurate, reliable, low-cost alternative to COP for balance assessment, particularly for Tandem stance. Furthermore, it provides the opportunity to take balance assessment further by providing the option of manipulating the visual field to induce changes in postural stability. This means balance assessments can probe more deeply the origins of any postural control deficits that may have been observed.

In terms of limitations, it is important to note that the VR technology can only measure movement of the head and hand controllers. This means using it to assess postural control during Single leg stance is limited as the sway data would not capture any balance errors associated with dropping the position of the non-supporting leg. New peripheral sensors that can be attached to the feet, but that are compatible with the Oculus Quest, will help overcome this problem in the future. Another limitation of this study is that special software is required to capture the movement data from the three controllers in real time to calculate total sway. The software used in this study (MOViR, INCISIV Ltd.) also generated the changes in visual field allowing for an accurate calculation of sway with respect to the visual condition experienced. Going forward it will be important that appropriate software for capturing movement data and manipulating the visual field is readily available to maximize the potential of using the Oculus Quest VR technology to assess balance.

In terms of applications, one of the other most striking areas where this type of assessment could have a large impact is on concussion detection and management. Current protocols rely heavily on human observation, meaning the subtle changes in balance that occur following a head injury may be missed. As Santos and colleagues [22] pointed out, using technology that is precise and reliable means concussion management can now be taken out of the lab and brought to the pitch. Having access to this type of technology and standardized tests would revolutionize the management of concussed players, allowing subtle changes in balance to be spotted, reducing the risk of having players return to play too quickly and putting themselves at increased risk of another injury. In fact, previous research has shown how balance abnormalities can be indicative of future risk of injury [23], but also a way to spot weaknesses that may directly impact on a player’s ability to execute a skill and perform effectively [4].

Like research a decade ago that showed how the Nintendo Wii balance board could be used to measure and train balance [5], this study shows how the Oculus Quest, a low-cost VR gaming headset can also be used to measure and train balance in older adults [24]. Previous research has shown how lab-based VR is well tolerated by more vulnerable groups, with studies showing how VR can be effectively used to cue gait in people with Parkinson’s [25,26], but also understand older adults’ decisions about when and how to cross a virtual road [27]. Given the power of this technology, creating VR balance games that use AI to adapt to the user’s abilities, opens a whole new vista in terms of balance rehabilitation and training possibilities [28,29]. Unlike the Nintendo Wii, where the parent company disinvested in the technology, multinational companies like Meta (Oculus Quest) and Bytedance (Pico Neo) are investing heavily in low-cost virtual reality hardware and relevant applications that will transform how we live our lives.

## 5. Conclusions

In conclusion, this study demonstrates how the Oculus Quest, a low-cost VR headset, can be used to reliably measure, but also challenge, a person’s ability to maintain their balance. Given the importance of postural control for a wide range of clinical applications, this technology offers promising new possibilities for not only balance assessment but also balance training.

## Figures and Tables

**Figure 1 diagnostics-12-01409-f001:**
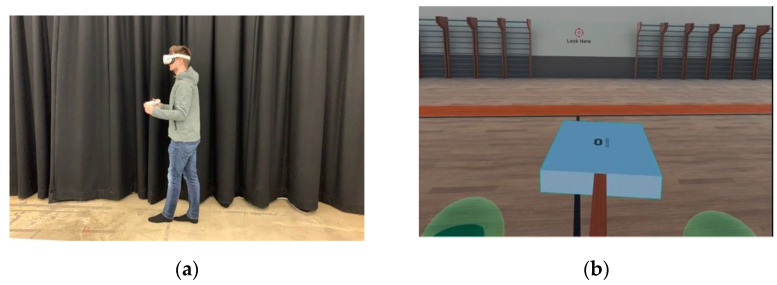
(**a**) A participant wearing the Oculus Quest headset and holding the two hand controllers. Any movements of the head and hands are captured by the motion controllers at 50 Hz. (**b**) An image showing what the participant saw inside the Oculus Quest headset when they were subjected to the ‘tilt’ visual field manipulation. Note the virtual green spheres at the bottom of the image which acted as a standardized visual reference for the hand position.

**Figure 2 diagnostics-12-01409-f002:**
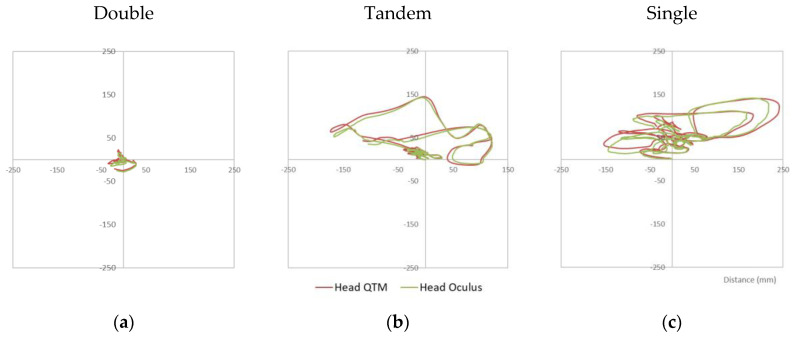
Three stabilogram plots representing the head sway (distance in mm) captured by the Oculus Quest (green) and the Qualisys motion capture cameras (red) in the three different stances (Double (**a**), Tandem (**b**), Single (**c**)) in two different axes (AP and ML).

**Figure 3 diagnostics-12-01409-f003:**
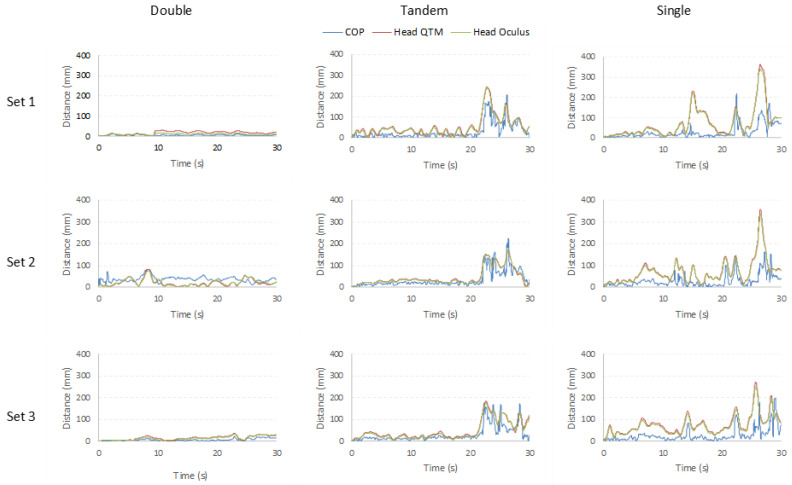
Graphs showing the distance covered by each of the three signals calculated in all three data sets for the COP (blue), Head Sway Qualisys (red) and Head Sway Oculus (green) measures. Notice how the correspondence is closest for all 3 signals in the Tandem stance data. This is also reflected in the ICC values presented in Table 2.

**Figure 4 diagnostics-12-01409-f004:**
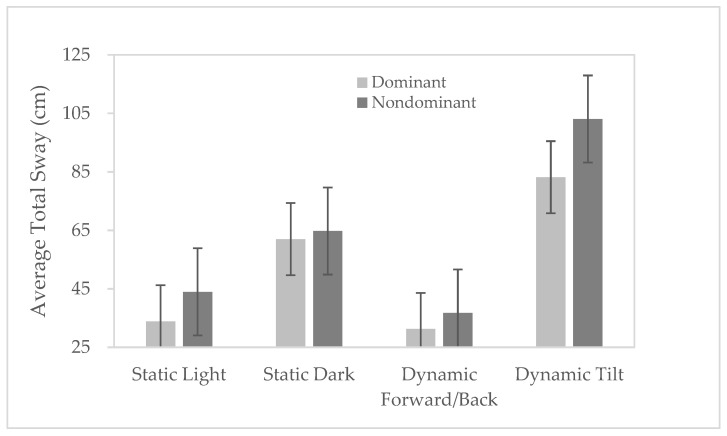
A graph showing the differences in mean total sway for the four different visual field conditions and for the dominant (light grey) and non-dominant (dark grey) stances. The error bars represent the standard deviations for the two different stance conditions.

**Table 1 diagnostics-12-01409-t001:** The average ICC and RMSE values for the Head movements recorded from the Qualisys and the Oculus Quest systems. The values are for the three axes (ML, AP and Vertical) and the calculated distance (sway) are presented for the three different stances (Double, Tandem, Single). ICC values can be interpreted as follows: ICC > 0.8 is excellent; ICC < 0.8 > 0.6 is good; ICC < 0.6 > 0.4 is moderate, while ICC < 0.4 is poor [19]. RMSE values are measured in mm with values closest to 0.0 indicating the highest levels of precision.

	Oculus vs. Qualisys Head Movement							
Stance	ICC				RMSE (mm)			
	ML	AP	Vertical	Distance	ML	AP	Vertical	Distance
Double	0.877	0.983	0.978	0.936	4.2	1.3	3.3	3.8
Tandem	0.994	0.996	0.937	0.994	3.4	4.2	1.6	3.9
Single	0.990	0.998	0.994	0.998	3.9	4.3	4.4	4.7

**Table 2 diagnostics-12-01409-t002:** The ICC values for head sway and CoP data in the ML, AP axes of motion for each of the three stances (Double, Tandem, Single). The ICC values for COP excursion and Head Sway (distance column) combine movement in both axes.

Stance	Oculus Head vs. COP (ICC)			Qualisys Head vs. COP (ICC)		
	ML	AP	Distance	ML	AP	Distance
Double	0.875	0.765	0.546	0.782	0.720	0.451
Tandem	0.687	0.858	0.888	0.697	0.850	0.879
Single	0.667	0.658	0.654	0.685	0.642	0.643

## Data Availability

Data are available at the following GitHub repository.

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
