# Peer review of "Can We Use the Oculus Quest VR Headset and Controllers to Reliably Assess Balance Stability?"

_diagnostics, 2022, doi:10.3390/diagnostics12061409_

Round 1

Reviewer 1 Report

 Thank you for the opportunity to re-review this manuscript. The authors have appropriately increased the number of participants to their model, providing a model that can be used to analyze balance perturbations. This is an innovative idea and is likely going to be helpful in clinical studies pending further research with patients with gait disorders of various etiology.

Author Response

We thank the reviewer for their careful consideration of our manuscript.

Reviewer 2 Report

No comments

Author Response

We thank Reviewer 2 for taking the time to review our manuscript.

Reviewer 3 Report

I have read the previous version of the article and it has been significantly improved.

In the discussion section, a paragraph is needed to acknowledge the limitations of the research.

Author Response

We thank Reviewer 3 for their careful consideration of our manuscript and kind comments. We have now added in a paragraph in the discussion around limitations of the study.

This manuscript is a resubmission of an earlier submission. The following is a list of the peer review reports and author responses from that submission.

Round 1

Reviewer 1 Report

Thank you for the opportunity to review this manuscript. This is an interesting concept. However, the methodology and analysis is preliminary and incomplete at best. The authors don't have a defined study population. Do they want to analyze gait and balance disorder of cerebellar ataxia/Parkinson disease/sensory ataxia/post concussive syndrome etc.. I got a sense that they were trying to present this analyses for healthy sportsmen but they make overarching claims about the utility of their analyses in older adults and various clinical scenarios. Their data is from one healthy volunteer. This study cannot even present if they can detect balance issue in that subject. This does not even qualify for n=1 study which would have been the case if she was followed longitudinally and developed gait issue that were compared to her initial analyses.

The presentation here in essentially a concept and not proof of concept. This does not meet the bar for publication in any reputed peer-reviewed journal. The authors need to recruit more participants and design a better study with a defined study population and comparison groups. This criticism is not to dissuade from any future work as this is a very innovative idea. But as it stands, their data is of no clinical value in my opinion.

Reviewer 2 Report

The article presents a timely and interesting research regarding a low-cost head-mounted display, the Oculus Quest.

The methodology and the results are well-presented.

While the references are is in the style of the journal, the reviewer believes that a few more citations should be added to the introductory section to create a better scientific background for the article.

Also, perhaps the Oculus Quest should be indicated in the title. Mentioning only low-cost head-mounted displays could mean several other devices.

The English used requires only a minor spellcheck.

Reviewer 3 Report

From the moment it became possible to study the balance in the vertical stance by instrumental methods, their availability and ease of use remain in the focus of attention of both scientists and practitioners.

For this reason, this work "Can we use a low-cost virtual reality headset to reliably assess 2 balance stability?" is relevant but not only as a method of registration. VR technology also allows for a specific impact, which also helps in testing the balance control system. Such testing cannot be carried out only through forceplatform. Now this is done by much more complex systems that implement the sensory organization test. The cost of any special system used today for carrying out balance testing that is close in meaning is orders of magnitude higher than the cost of the VR system used in the work. If clinicians could have a simple and inexpensive tool at their disposal, the implications for identifying and treating pathologies where balance is essential cannot be overestimated. So this study is a very good step in the right direction. However, there is a pitfall. The industry, in many ways, determines the opportunities for both practitioners and researchers. In my humble experience, the use of consumer devices, like VR helmets or the Nintendo Wii quoted in the paper, cannot become practical medicine tools. This requires specialized devices. However, for the study of the principles themselves, they are quite applicable, which was brilliantly shown by the authors.

I cannot insist on this, but it seems to me that one photograph of the object of study during the registration could decorate the work.